# WHICH LLM TO PICK?
# ONLINE ACTIVE MODEL SELECTION FOR
# LARGE LANGUAGE MODELS

## ABSTRACT

Large Language Models (LLMs) are increasingly applied to process streaming data, with practitioners relying on benchmarks to select the best model even though these signals only approximate real performance. While oracle annotations can provide reliable feedback, they are often costly and difficult to obtain at scale. To address this challenge, we propose ONLINE LLM PICKER, the first framework for active model selection for LLMs in online settings. Given an arbitrary stream of queries and a limited annotation budget, ONLINE LLM PICKER selects the most informative prompts for annotation to identify the best LLM among candidate models. Across multiple tasks including 10 datasets, for over 130 language models, we show that ONLINE LLM PICKER saves annotation cost by up to 71.67% while reliably identifying the best or near-best model for the stream. We also show that using the returned model for sequential generation on unannotated prompts across the stream reduces regret by up to a factor of $2.51\times$, indicating that ONLINE LLM PICKER can identify the best or near-best model well before processing all streaming prompts.

## 1 INTRODUCTION

Large Language Models (LLMs) are widely applied across various domains (Fan et al., 2023; Tan, 2023), from medicine (Boll et al., 2025; Veen et al., 2024; Biswas & Talukdar, 2024) and industry (Angelopoulos et al., 2025; Kok et al., 2024; Li et al., 2024b), to education (Dan et al., 2023; Caines et al., 2023) and law (Pont et al., 2023; Lee, 2023), (Guha et al., 2023). Alongside this growing adoption, hundreds of publicly available LLMs have emerged (Google-t5, 2024; Meta-Llama, 2024; Falcon, 2023; DeepSeek, 2025; MistralAI; Qwen, 2025; Nvidia, 2025), many demonstrating strong performance without the need for fine-tuning and the ability to follow instructions and adapt to new tasks with little or no task-specific supervision (Brown et al., 2020; Kojima et al., 2023; Dong et al., 2024; Liu et al., 2021).

Given the abundance of off-the-shelf models, selecting the most suitable LLM for a particular task or data stream is a non-trivial problem. Existing evaluation pipelines and benchmarks assess performance across diverse datasets and tasks using multiple metrics (Guo et al., 2023), but no single model consistently excels across domains or datasets (Chang et al., 2023; Liang et al., 2023). Because LLM effectiveness is highly context-dependent and varies substantially across scenarios, standard evaluation metrics alone cannot fully capture model quality or practical utility (Ouyang et al., 2022; Kocoń et al., 2023). A common approach to model selection is therefore to rely on randomly or heuristically chosen small subsets of annotated data (Polo et al., 2024; Vivek et al., 2024). However, such strategies often lead to inefficient use of resources and fail to reliably capture differences across models (Kossen et al., 2021). Active Model Selection (Madani et al., 2012; Karimi et al., 2021; Ashury-Tahan et al., 2024; Okanovic et al., 2024; Kay et al., 2025; Liu et al., 2022; Li et al., 2024a; Hara et al., 2024; Gardner et al., 2015) addresses this limitation by selectively annotating a small set of queries to identify the best model for arbitrary data examples. Yet, prior work has largely focused on classification tasks rather than generation (Karimi et al., 2021; Liu et al., 2022; Okanovic et al., 2024; Kay et al., 2025; Li et al., 2024a; Hara et al., 2024), with the exception of Ashury-Tahan et al. (2024), which studies language model selection under the assumption that all

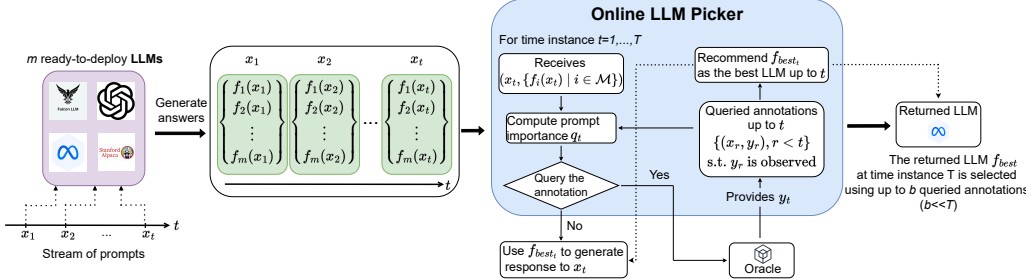

Figure 1: An overview of ONLINE LLM PICKER workflow.

prompts are available from the outset. To the best of our knowledge, no prior work has explored active model selection for LLMs in online settings to date.

**Contributions**: To fill this gap, in this work we propose ONLINE LLM PICKER, an active model selection strategy designed to efficiently identify the best candidate LLM in online settings. Given an incoming stream of prompts and a small annotation budget, ONLINE LLM PICKER selects the most informative prompts to annotate in order to reliably identify the best LLM for the stream. Our framework is based on the combined informativeness criterion that maximizes variance across model responses to unannotated prompts while also accounting for posterior uncertainty. ONLINE LLM PICKER makes no assumptions about the underlying LLMs, allowing it to be applied to any generative task and any model collection. ONLINE LLM PICKER is depicted in Figure 1.

We validate ONLINE LLM PICKER through a comprehensive set of experiments spanning multiple LLM generation tasks, including open-ended question answering (Rajpurkar et al., 2016), open-ended question answering with no correct response (Rajpurkar et al., 2018), grammar correction of natural language text (Loem et al., 2023), and both arithmetic (Gambardella et al., 2024) and calculus problems expressed in code-like notation (Gao et al., 2023). Our model collections across these tasks contain over 130 language models. We rank the models using both ROUGE (Lin, 2004) and embedding-based similarity measures (e.g. BERTScore (Zhang et al., 2020)), so that we can capture lexical and semantic agreement with the reference outputs. Our results demonstrate that ONLINE LLM PICKER identifies the best or near-best LLM in online environments while requiring up to 71.67% fewer annotations than competing baselines, and reduces regret by up to a factor of $2.51\times$ when applied to sequential generation on unannotated prompts across the stream.

## 2 RELATED WORK

A wide range of metrics are proposed for the **evaluation of language models**, from early automatic metrics such as ROUGE (Lin, 2004), BLEU (Papineni et al., 2002), and METEOR (Lavie & Agarwal, 2007) to benchmark-driven progress with GLUE (Wang et al., 2019), SuperGLUE (Wang et al., 2020), MMLU (Hendrycks et al., 2021), and BIG-bench (Srivastava et al., 2023). More recently, evaluation has shifted toward human feedback (Stiennon et al., 2022); (Ouyang et al., 2022), but reliance on annotators is costly and limits scalability in large-scale or streaming settings. LLM-as-a-judge methods (Gu et al. (2025); Bai et al. (2024); Chiang et al. (2024); Dubois et al. (2025)) offer more scalable pipelines, though they introduce bias. Efficiency is also identified as a central concern with Active Testing (Berrada et al. (2025)) which reduces annotation cost of evaluation through adaptive sampling. However, such methods focus on evaluating a fixed model or small set of models, whereas our setting requires identifying the best LLM for an arbitrary data stream under limited annotations.

The most relevant line of work for our setting is **active model selection**, which seeks to identify the best model from a set of candidates by querying informative examples (Madani et al., 2012; Karimi et al., 2021; Ashury-Tahan et al., 2024; Okanovic et al., 2024; Kay et al., 2025; Liu et al., 2022; Li et al., 2024a; Hara et al., 2024; Gardner et al., 2015) serving as the primary motivation for our work. However, most of these approaches assume a pool-based setup where all queries are available upfront(Okanovic et al., 2024; Ashury-Tahan et al., 2024; Kay et al., 2025) , or focus on

classification tasks(Karimi et al., 2021; Liu et al., 2022; Okanovic et al., 2024; Kay et al., 2025; Li et al., 2024a; Hara et al., 2024) , and therefore not directly applicable to our setting.

In this work, we introduce the novel problem of online active model selection for large language models and present several strategies as baselines (introduced in Section 4) along with ours. We relegate discussions of related but less directly relevant frameworks to Appendix A.

# 3 ONLINE LLM PICKER

## 3.1 PROBLEM STATEMENT AND BACKGROUND

Consider the inference phase with a stream of $T$ prompts $\{x_t \in \mathcal{X} \mid t \in [T]\}$ drawn from an unknown distribution. Each prompt $x_t$ is associated with a reference annotation $y_t \in \mathcal{Y}$, which remains hidden unless queried from an oracle.

We consider $m$ pretrained language models $\mathcal{M} = \{f_i : \mathcal{X} \to \mathcal{Y} \mid i \in [m]\}$. For each prompt $x_t$, model $i$ produces a response $f_i(x_t)$. At time step $t$, given the set of model responses $\{f_i(x_t) \mid i \in \mathcal{M}\}$, we decide whether to query the oracle for the reference annotation $y_t$ or not. If $y_t$ is not queried, it remains hidden; otherwise, we compute the loss vector as

$$\boldsymbol{\ell}_t^y = [\ell_{t,i}^y]_{i \in \mathcal{M}} \quad \text{where} \quad \ell_{t,i}^y = 1 - d\big(y_t, f_i(x_t)\big) \tag{1}$$

for some similarity score $d : \mathcal{Y} \mapsto \mathcal{Y}$. In our framework, we choose $d(\cdot, \cdot)$ to be ROUGE-L (Lin, 2004) or BERTScore (Zhang et al., 2020) in our case (Rehman et al., 2025; Sul & Choi, 2023).

Given an annotation budget $b \ll T$ for the entire stream, our objective is to select the best language model for the remaining $T - b$ prompts using only the annotated prompts. At each time step $t$, our method returns the language model $f_{\text{best}_t}$ based on the annotations observed up to time $t$ such that $f_{\text{best}_t} \sim \mathbf{p}_t$. Our objective is to annotate the most informative examples across the stream so that $f_{\text{best}_t}$ closely approximates the *true* best language model, defined as the model that incurs the minimum cumulative loss if all annotations in the stream were available: $f_{\text{best}} := \arg\min_{i \in \mathcal{M}} \sum_{t=1}^{T} \ell_{t,i}^y$. Formally, we aim to minimize the cumulative loss of $f_{\text{best}_t}$ relative to $f_{\text{best}}$, which is captured by the standard regret:

$$R_T = \sum_{t=1}^{T} \ell_{t, f_{\text{best}_t}}^y - \sum_{t=1}^{T} \ell_{t, f_{\text{best}}}^y. \tag{2}$$

Since our ultimate goal is to identify the best model under the annotation budget $b$, we also evaluate the quality of the returned model using two complementary metrics. The first is *identification probability*, which measures the probability of successfully recovering the true best model in a stochastic data stream. The second is *annotation efficiency*, which quantifies the annotation cost saved by identifying the best or near-best models under budget $b$. We provide their formal definitions in Section 4.

## 3.2 THE ALGORITHM

In this section, we introduce ONLINE LLM PICKER, which aims to identify the best language model for a given stream of prompts with limited reference annotations from the oracle.

At a high level, our algorithm operates as follows. At each time step $t$, we decide whether to query the annotation of the prompt $x_t$ via a random experiment: we draw a Bernoulli random variable $Q_t \sim$ Bernoulli($q_t$), with *query probability* $q_t$, where the choice of $q_t$ depends on factors introduced later. If a query is made ($Q_t = 1$), we update our posterior belief over the models using the newly observed annotation. We denote this posterior by $\mathbf{p}_t = [p_{t,i}]_{i \in \mathcal{M}}$ where $p_{t,i}$ represents the probability each model $i$ at time $t$ being the true best language model $f_{\text{best}}$. If no query is made ($Q_t = 0$), the reference annotation for $x_t$ remains hidden and the posterior remains unchanged.

To perform the posterior update, we first construct an importance-weighted loss estimate using $\ell_{t,i}^y$ in Equation 1:

$$\hat{\ell}_{t,i}^y = \frac{\ell_{t,i}^y}{q_t} Q_t \quad \forall i \in \mathcal{M}$$

where $1/q_t$ serves as an importance weight, correcting sampling bias from the querying process by assigning greater weight to under-sampled instances.

We then obtain the cumulative loss estimate up to time $t$ as

$$\hat{L}_{t,i} = \hat{L}_{t-1,i} + \hat{\ell}_{t,i}^y \quad \forall i \in \mathcal{M}$$

and update the posterior distribution $\mathbf{p}_t = [p_{t,i}]_{i \in \mathcal{M}}$ using the Exponential Weights (EW) algorithm by Littlestone & Warmuth (1994) with an adaptive learning rate $\eta_t$:

$$p_{t,i} \propto \exp\{-\eta_t \hat{L}_{t-1,i}\}. \tag{3}$$

Appendix E details the derivation of $\boldsymbol{p}_t$. We expand on our learning rate $\eta_t$ as follows.

### 3.2.1 ADAPTIVE LEARNING RATE

While time-based decay schedules are common (Karimi et al., 2021), inspired by AdaHedge (Rooij et al., 2013), we propose an adaptive learning rate $\eta_t$ that updates based on observed annotations. In AdaHedge, the dynamic learning rate is defined in terms of the cumulative mixability gap, which measures the cumulative approximation error incurred when approximating the Hedge loss with the mix loss (Rooij et al., 2013). By means of Bernstein's bound, the mixability gap at a certain round $t$ can be expressed in terms of variance of the losses. In addition, Rooij et al. (2013) shows that the cumulative mixability gap grows at most as the square root of the cumulative loss variance up to $t$. Motivated by this, we introduce an adaptive learning rate that depends on the variance of the losses over the queried examples. The variance of the loss can be computed as:

$$\operatorname*{Var}_{i \sim \boldsymbol{p}_t} \ell_{t,i}^y = \langle \boldsymbol{p}_t, \boldsymbol{\ell}_t^y \odot \boldsymbol{\ell}_t^y \rangle - (\langle \boldsymbol{p}_t, \boldsymbol{\ell}_t^y \rangle)^2.^1 \tag{4}$$

Although variance-based strategies exist for adapting the learning rate (Rooij et al., 2013), they are less effective with limited annotations, where sparse queries and a few noisy examples can disproportionately distort the variance estimate. Following the idea of exponential moving averages in optimization (Kingma & Ba, 2017), which keeps exponential moving averages of the gradient and its square (first and second moments), applies bias correction, and uses them to form a per-parameter adaptive step, we instead introduce an exponential moving average of the variance of the observed examples as a proxy. This proxy does not track the exact variance of the underlying losses, but provides a smoothed, adaptive estimate that emphasizes recent observations while discounting stale ones, thus makes the learning rate more robust under sparse and noisy feedback.

We compute the exponential moving average of the variance as:

$$\mathcal{V}_t = \begin{cases} \beta \cdot \mathcal{V}_{t-1} + (1 - \beta) \cdot \operatorname*{Var}_{i \sim \boldsymbol{p}_t} \ell_{t,i}^y, & \text{if } y_t \text{ is observed,} \\ \mathcal{V}_{t-1}, & \text{otherwise,} \end{cases} \tag{5}$$

where $\beta \in [0, 1]$ is the decay rate of the exponential moving average. The effective window size of this exponential moving average is approximately $(1 - \beta)^{-1}$, meaning that values of $\beta \approx 1$ place more weight on past observations and correspond to a longer memory.

At the beginning of the stream, we have $\mathcal{V}_0 = 0$ and no annotations are available yet. This initialization biases the variance proxy toward zero, particularly in the early time instances. As discussed in Kingma & Ba (2017), this bias can be corrected by a bias-adjusted proxy of the average variance:

$$\hat{\mathcal{V}}_t = \frac{\mathcal{V}_t}{1 - \beta^t}. \tag{6}$$

To connect back to the framework of Rooij et al. (2013), which relates the cumulative mixability gap to the cumulative variance, we estimate the cumulative variance up to time $t$ by scaling the number of time steps with the bias-adjusted proxy of the average variance with $t \cdot \hat{\mathcal{V}}_t$ (6). Eventually, at any time instance $t$, we compute the adaptive learning rate $\eta_t$ as follows:

$$\eta_t = \sqrt{\frac{\log m}{t \cdot (\hat{\mathcal{V}}_t + \epsilon)}} \tag{7}$$

---

[1] Here, $\odot$ denotes the element-wise product and $\langle \cdot, \cdot \rangle$ denotes the inner product.

where $\epsilon > 0$ is a small constant added for numerical stability. This makes sure that $\eta_t$ remains finite and well-defined for all $t$, including the case $\hat{\mathcal{V}}_t \to 0$, which corresponds to (near-) identical pairwise losses among the answers generated by language models. With a uniform prior over the $m$ experts, the initial information cost of not knowing the best expert is $\log m$, which appears as $(\log m)/\eta$ in the regret bound (Rooij et al., 2013).

Plugging Equation 7 into Equation 3 completes the model posterior update. We now discuss the query probability $q_t$ (10), which determines the outcome of the query decision at each time step $t$.

### 3.2.2 QUERY PROBABILITY

The query probability $q_t$ at each time step is designed to capture our uncertainty regarding the identity of the best model on the given prompt $x_t$, and subsequently guide our decision of whether to request the annotation of $x_t$ or not. Importantly, this probability should not only reflect predictive uncertainty but also integrate the current posterior belief over the language models, and balance the competing objectives of exploration and exploitation by doing so. A natural way to operationalize uncertainty in streaming settings is through variance-based measures(Karimi et al., 2021), which serve as a principled proxy for the potential information gain associated with querying an annotation. Intuitively, the goal is to estimate the expected value of acquiring the annotation for $x_t$, that is, the degree to which it would refine our estimate of the best model. More concretely, this quantity can be estimated from the losses incurred by the model responses $\{f_i(x_t) \mid i \in \mathcal{M}\}$, together with the posterior distribution over models $\boldsymbol{p}_t$ at time $t$. Since the reference annotation $y_t$ is not available, this variance estimation is hypothetical. In the classification setting, such a hypothetical variance is well-defined because the true label at time $t$ is assumed to lie within a finite set of predefined classes. In contrast, in the generative setting there is no fixed label space: the set of valid responses is open-ended and potentially unbounded, which makes evaluation and comparison substantially more challenging. To address this, we approximate the response space at each time $t$ by considering the complete set of language model outputs $\{f_i(x_t) \mid i \in \mathcal{M}\}$.

Towards that, we denote the (pairwise) loss between the response generated by the language models and that generated by the model $k$ by $\boldsymbol{\ell}_t^k = [\ell_{t,i}^k]_{i \in \mathcal{M}}$ where $\ell_{t,i}^k = 1 - d(f_i(x_t), f_k(x_t))$. At each time instance $t$, we treat the response of each language model $k \in \mathcal{M}$ as the hypothetical reference annotation. We then compute the maximum hypothetical variance among the losses with:

$$\max_{k \in \mathcal{M}} \operatorname{Var}_{i \sim \boldsymbol{p}_t} \ell_{t,i}^k = \langle \boldsymbol{p}_t, \boldsymbol{\ell}_t^k \odot \boldsymbol{\ell}_t^k \rangle - (\langle \boldsymbol{p}_t, \boldsymbol{\ell}_t^k \rangle)^2 \tag{8}$$

which represents our uncertainty about the annotation of $x_i$ given the model posterior $\boldsymbol{p}_t$. If the language model responses are nearly identical where $\boldsymbol{\ell}_t^k \approx 0$, or if they differ to the same degree such that $\boldsymbol{\ell}_t^k$ is similar for all $k \in \mathcal{M}$, then the dispersion under $\boldsymbol{p}_t$ is negligible. In this case, the annotation $y_t$ provides little information, contributes minimally to model selection, and has only a negligible effect on regret. Conversely, if the maximum hypothetical variance is large, for instance, when $\boldsymbol{\ell}_t^k$ exhibits substantial heterogeneity across model responses, then the annotation $y_t$ is informative.

Revisiting the model posterior, which our algorithm updates based on the observed annotations, we find that in some cases it concentrates around a single best model, whereas in others it remains diffuse across multiple candidates. When the posterior distribution is already concentrated after some annotations, additional annotations have very little impact on the posterior update. Otherwise, a few strategically chosen annotations can shift the balance and determine the best model. This perspective is rooted in the Bayesian experimental design, where the value of new information measured by its expected reduction in posterior uncertainty. Shannon entropy of the posterior provides a natural quantification of 'how much remains to be learned' about which model or parameter is best (Lindley, 1956; MacKay, 1992; Sebastiani & Wynn, 2025). A parallel also exists in online decision making: exploration is most valuable when the uncertainty expressed in terms of posterior entropy about the best choice is high, since this is when information can most effectively reduce future regret (Russo & Roy, 2016; 2017).

Combining multiple signals such as variance and entropy is a common design pattern in active learning and streaming settings, such as scaling committee disagreement by input density (McCallum & Nigam, 1998), weighting predictive entropy by local density (Zhu et al., 2008), querying when both uncertainty and density are high (Ienco et al., 2014), and multiplying predictive entropy by

a coverage-based representativeness factor (Katragadda et al., 2023). Following this principle, we incorporate posterior entropy into our query probability. Specifically, we compute the normalized entropy of the posterior distribution over models:

$$\bar{\mathbb{H}}(\boldsymbol{p}_t) \ = \ \frac{\mathbb{H}(\boldsymbol{p}_t)}{\log m} \in [0, 1] \qquad \text{where} \ \ \mathbb{H}(\boldsymbol{p}_t) \ = \ -\sum_{i=1}^{m} p_{t,i} \log p_{t,i}. \tag{9}$$

We normalize entropy by $\log m$ to place uncertainty on a fixed $[0, 1]$ scale, independent of the number of models $m$. This makes thresholds and query schedules comparable across settings and interpretable: $\bar{\mathbb{H}} = 0$ when one model dominates (high confidence) and $\bar{\mathbb{H}} = 1$ under a uniform posterior (maximal uncertainty). We then incorporate this normalized entropy by scaling the variance in 8 such that $\underset{i \sim \boldsymbol{p}_t}{\text{Var}} \ell_{t,i}^k \cdot \bar{\mathbb{H}}(\boldsymbol{p}_t)$.

Finally, we define the query probability as

$$q_t = \begin{cases} \max\left\{\max_{k \in \mathcal{M}} \underset{i \sim \boldsymbol{p}_t}{\text{Var}} \ell_{t,i}^k \cdot \bar{\mathbb{H}}(\boldsymbol{p}_t), \eta_t\right\}, & \text{if } \max_{k \in \mathcal{M}} \underset{i \sim \boldsymbol{p}_t}{\text{Var}} \ell_{t,i}^k \neq 0 \\ 0, & \text{otherwise.} \end{cases} \tag{10}$$

where $\eta_t$ is a time-decaying lower bound based on the adaptive learning rate. This bound prevents two problems: when variance is extremely small, importance-weighted loss estimates can become unstable and inflate regret; and when predictions are overly confident, the algorithm may skip prompts that are actually informative.

In essence, the query rule is based on the hypothetical loss variance across models under the current posterior as well as the direct posterior entropy. High variance and entropy trigger more queries, while a concentrated posterior reduces them. Early on, queries are mainly variance-driven, but over time $q_t$ balances both factors and reflects model disagreement and overall uncertainty about the annotation of the incoming queries.

The pseudocode of ONLINE LLM PICKER is depicted in Algorithm 1.

---

**Algorithm 1** Online LLM Picker

**Require:** Language models $\mathcal{M}$

    Set $\hat{L}_{0,i} = 0 \quad \forall i \in \mathcal{M}, \mathcal{V}_0 = 0$

    **for** $t = 1, 2, ..., T$ **do**

        $\eta_t = \sqrt{\frac{\log m}{t \cdot (\mathcal{V}_t + \epsilon)}}$

        $p_{t,i} \propto \exp\{-\eta_t \hat{L}_{t-1,i}\} \quad \forall i \in \mathcal{M}$              ▷ update posterior over experts

        Get models predictions $\{f_i(x_t) \mid i \in \mathcal{M}\}$ for $x_t$

        Recommend $f_{\text{best}_t} := \arg\max_{i \in \mathcal{M}} p_{t,i}$ as the best model up to time instance $t$

        Compute $q_t$ as in *(10)* and sample $Q_t \sim \text{Bernoulli}(q_t)$

        **if** Q=1 **then**                                  ▷ query the annotation $y_t$

            Update $\mathcal{V}_t$ as in *(5)*

            $\hat{L}_{t,i} = \hat{L}_{t-1,i} + \frac{\ell_{t,i}^y}{q_t} \quad \forall i \in \mathcal{M}$

        **else**

            Annotate $x_t$ with $f_{\text{best}_t}(x_i)$

            $\hat{L}_{t,i} = \hat{L}_{t-1,i}$

        **end if**

    **end for**

**Return:** $f_{\text{best}} = \arg\max_{i \in \mathcal{M}} p_{i,T}$             ▷ return the best model for the stream

---

## 4 EXPERIMENTS

We evaluate ONLINE LLM PICKER for active LLM selection in streaming settings using several public generative datasets and model collections containing more than 130 language models. As this is the first study of its kind, we introduce baselines and compare performance across multiple

metrics: *identification probability*, *regret*, and *annotation efficiency* for identifying best or near-best models across the stream.

## 4.1 DATASETS AND MODELS

We benchmark ONLINE LLM PICKER against competing baselines across datasets from different generative tasks. First, we test the algorithms on LLMs collections for the SQuAD (Rajpurkar et al., 2016) and SQuAD v2 (Rajpurkar et al., 2018) datasets, which are designed for question answering tasks, with the latter including questions that may not have an answer in the given context. Question answering is often framed as a non-generative task, especially in the extractive setting where models directly copy answer spans from the input text. In contrast, our models generate answers in their own words, producing open-ended text instead of directly extracting it. We also use Comprehensive Arithmetic Problems dataset (Lee, 2024) that features a range of algebraic expressions, the Calculus Datasets (FDU) (Zhang, 2025) that consist of more advanced numerical and symbolic problems represented in LaTeX code, and the Grammar Correction dataset (Agentlans, 2024), which challenges models to fix grammar mistakes in English sentences. In addition, we also consider MT-Bench (Bai et al., 2024), a benchmark for evaluating multi-turn conversational ability of LLMs, and multiple datasets from the HELM Benchmark(Liang et al., 2023). In particular, we include two Longform Question Answering (LF-QA) from classic HELM, LF-QA Canonical and LF-QA Prompt, respectively. From MedHELM (Bedi et al., 2025) we add MedCalc (Khandekar et al., 2024) to our experiments. We also include FinQA(Chen et al., 2022) from HELM Finance.

As for the language models, we include several open-source and proprietary LLMs (Almazrouei et al. (2023), Radford et al. (2019), Brown et al. (2020), Raffel et al. (2023), OpenAI et al. (2024), DeepSeek-AI et al. (2025), Touvron et al. (2023), Grattafiori et al. (2024)), ranging from those with a few million parameters to those with several billion. Some of these models are fine-tuned on the specific datasets of interest, while others are general-purpose models suitable for a wide range of generative tasks. We also expand our model collection to DeepSeek (DeepSeek-AI et al., 2025) and various GPT-based models (Brown et al., 2020; Kocoń et al., 2023; Ouyang et al., 2022). An overview of our datasets and models are in Table 2 as well as LLM scores on the test datasets in Figure 4 in Appendix B. To simulate multiple LLMs using the same underlying model, we also apply prompt engineering techniques to instruction-tuned LLMs, encouraging them to behave in agent-specific ways. Prompt engineering is also used to improve performance on our benchmark datasets by reducing unnecessary verbosity and avoiding irrelevant explanations in the output. Our work is model-agnostic and makes no assumptions about LLM architectures or performance. To approximate real-world use, we evaluate across a diverse suite of datasets. More details on LLM collections and datasets can be found in Appendix B.

## 4.2 BASELINES

We introduce several strategies and baselines and evaluate the performance of ONLINE LLM PICKER against them. These methods typically follow a coin-flipping strategy. At each time instance $t$, when a new prompt $x_t$ is received, the decision to query the reference annotation $y_t$ is made by sampling a Bernoulli random variable $Q_t$ with bias $q_t$, which is usually adaptive. The reference annotation $y_t$ is queried only if $Q_t = 1$.

For **Random** (passive learning) baseline, we query annotation of each time instance with a fixed probability $q_t = b/T$, having an expected number of $b$ annotations queried over a stream of length $T$. For **Disagreement** baseline, we only consider prompts where the language models strongly disagree and use a fixed $q_t = b/T$ on those instances. Disagreement is quantified as the variance across experts in their average pairwise discrepancies, and a query is triggered when this measure exceeds a small threshold we introduced. We adapt the **Kullback–Leibler** baseline where KL queries with probability proportional to the Kullback–Leibler divergence (Shlens, 2014) between distribution of pairwise losses at time instance $t$ and the posterior belief, to encourage annotations when the loss distribution deviates strongly from the learned belief. We adapt **Uncertainty**(Dagan & Engelson, 1995) to our setting to query annotations with probability similar to the Shannon entropy of the pairwise losses over candidate models. It estimates how evenly the models support competing hypotheses and regards higher entropy as greater uncertainty.

### 4.3 EVALUATION PROTOCOL

We evaluate ONLINE LLM PICKER separately on each dataset. For a given dataset, we generate a stream by drawing $T$ i.i.d. instances uniformly at random and feeding them to each method, referring to each such stream as a *realization*. For each realization, we evaluate performance based on the language model returned by each method. We repeat this process over multiple independent realizations with fresh streams drawn from the test set, and average the results to estimate the expected performance for each metric.

We define the budget $b$ as the maximum number of annotations that a method can query within a given realization. We evaluate the methods across different budget levels. For a fair comparison under the same budget, we tune the hyperparameter(s) of each method to query the same number of annotations in average at the end of the stream and compare their performance. Hyperparameter tuning adjusts the query probability of a method by multiplying it with different up- or down-scaling factors to control how many annotations each method ends up requesting in average across all realizations.

### 4.4 PERFORMANCE METRICS

We evaluate each algorithm using the following metrics. First, for a fixed annotation budget, we compute the *regret* to measure how well the models returned by each method perform in sequential generation on the unannotated prompts across the stream. Second, we measure *identification probability*, defined as the fraction of realizations in which the best model is correctly identified by the end of the stream. Finally, we assess *annotation efficiency*, which captures how efficiently a method identifies the best or near-best language models relative to the number of annotations used.

### 4.5 EXPERIMENTAL RESULTS

#### 4.5.1 REGRET

For a given budget where ONLINE LLM PICKER returns a model with high confidence, we report the expected regret averaged over all realizations. Our experiments are robust on the choice of budgets and we evaluate ONLINE LLM PICKER and baselines at very different budget levels. In all cases, ONLINE LLM PICKER consistently achieves a reduction in regret, in some cases reaching up to a factor of $2.51\times$ with respect to the best competing baseline, which shows its ability for sequential generation even well before exhausting its annotation budget. Details on annotation budget levels and further results are reported in Table 3 of Appendix D.

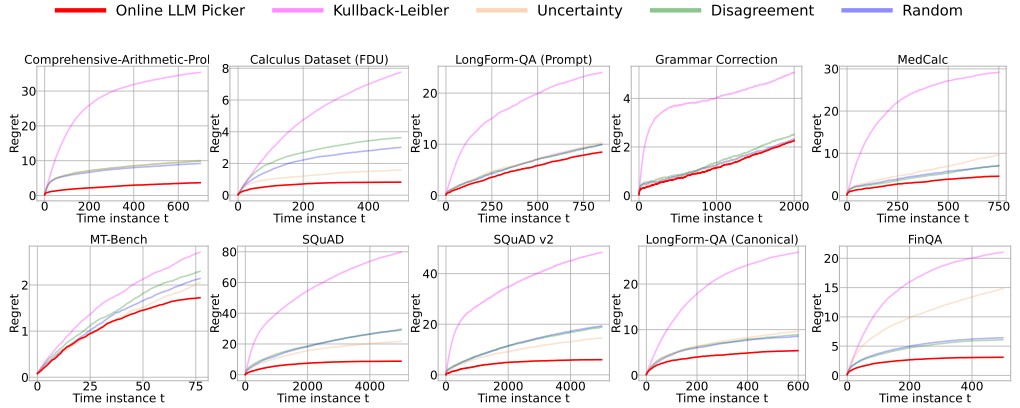

Figure 2: Regret for ONLINE LLM PICKER and baselines across 10 datasets.

#### 4.5.2 IDENTIFICATION PROBABILITY

Figure 3 shows the identification probabilities for ONLINE LLM PICKER and the baselines. For each dataset, we extend the annotation budget until the best competing baseline achieves $100\%$

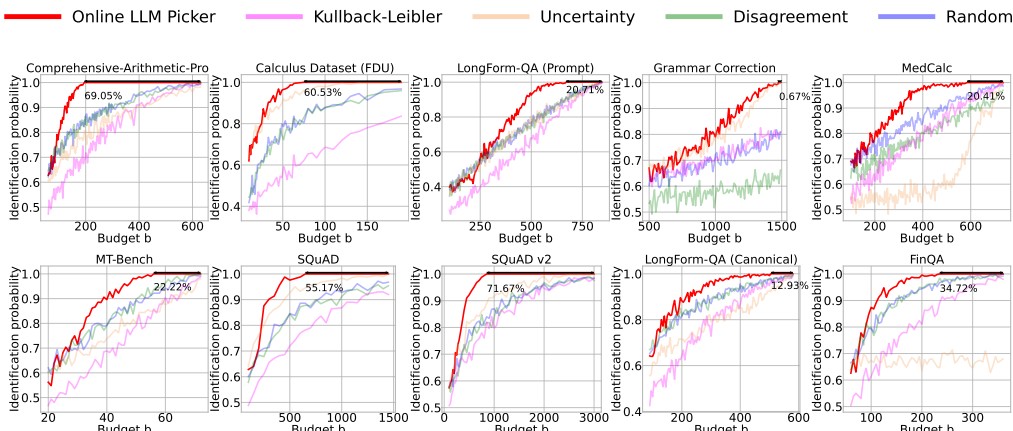

Figure 3: Identification probabilities of ONLINE LLM PICKER and baselines across different annotation budgets.

identification probability, and then report the percentage of annotation cost saved by ONLINE LLM PICKER for confidently identifying the best model. Across all datasets and annotation budgets, ONLINE LLM PICKER consistently requires the fewest annotations to reach maximum identification probability, showing its ability in correctly identifying the most informative examples to annotate, saving up to 71.67% with respect to the budget needed by the best competing baseline.

### 4.5.3 ANNOTATION EFFICIENCY

We examine the annotation cost of selecting a best or near-best model with accuracy within the $\delta$-vicinity of that of the true best model across the entire stream. In particular, we focus on the required number of annotations where, in all realizations, the selected models are within 2.5%, 1%, 0.5%, and 0.1% of the true best model score. We evaluate the percentage reduction in annotation cost relative to the best competing baseline, reported separately for each dataset and each $\delta$-vicinity. Our results in Table 1 show that ONLINE LLM PICKER significantly reduces up to 70.69% the number of annotations required to identify a near-best model whose accuracy is close to that of the best one. ONLINE LLM PICKER is consistently more annotation-efficient in identifying both the best and a near-best model across different tasks and datasets.

| Dataset | $\delta$=2.5% | $\delta$=1.0% | $\delta$=0.5% | $\delta$=0.1% |
|---|---|---|---|---|
| Comprehensive–Arithmetic–Problems | ↓ **24.39%** | ↓ **64.89%** | ↓ **68.00%** | ↓ **69.05%** |
| Calculus Dataset (FDU) | ↑ 28.00% | ↓ **53.75%** | ↓ **46.43%** | ↓ **60.53%** |
| LongForm–QA (Prompt) | 0.00% | ↓ **27.70%** | ↓ **25.79%** | ↓ **20.96%** |
| Grammar Correction | 0.00% | 0.00% | 0.00% | ↓ **0.67%** |
| MedCalc | ↓ **14.66%** | ↓ **38.46%** | ↓ **35.51%** | ↓ **21.09%** |
| MT–Bench | ↓ **22.22%** | ↓ **22.22%** | ↓ **22.22%** | ↓ **22.22%** |
| SQuAD | ↑ 20.00% | ↓ **52.00%** | ↓ **55.17%** | ↓ **55.17%** |
| SQuAD v2 | 0.00% | ↓ **33.33%** | ↓ **63.04%** | ↓ **70.69%** |
| LongForm-QA (Canonical) | ↓ **30.30%** | ↓ **27.00%** | ↓ **33.93%** | ↓ **23.28%** |
| FinQA | ↓ **34.69%** | ↓ **31.88%** | ↓ **34.72%** | ↓ **34.72%** |

Table 1: Annotation efficiency for the near-best model.

## 5 DISCUSSIONS

We introduce the novel problem of online model selection for language models with limited annotation evidence, which is a technically challenging setting given the open-ended nature of generative outputs. We propose ONLINE LLM PICKER, a method tailored to this task that confidently identifies the best language model for the task in an annotation-efficient manner. As LLMs are increasingly deployed in domains where annotation budgets are limited and data distributions shift, ONLINE LLM PICKER enables adaptive model selection that reduces annotation costs, maintains performance under evolving conditions, and improves the robustness of real-world LLM deployment.

**Ethics statement.** We do not foresee ethical concerns arising from this work. We introduce a novel active model selection algorithm that performs well under budget constraints and compare it with established baselines in the literature. The study uses public benchmarks and does not involve human subjects or sensitive data.

**Reproducibility statement.** Our results are completely reproducible. We present the rigorous workflow in the paper and report experiments in the main text and the appendix. We provide the complete source code and scripts as supplementary material to reproduce all experiments.

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

## A    RELATED WORK

**Data sampling methods for LLMs** Data sampling has long been an important topic in machine learning, and with the rise of LLMs it has gained renewed attention. Recent research explores diverse strategies for selecting and ordering training data to improve efficiency, scalability, and final model quality. A first branch of approaches follows *curriculum learning*(Bengio et al., 2009), where examples are ordered from easy to hard to smooth optimization. Zhang et al. (2025) ranks data incrementally by means of different metrics for Language Models pretraining, Fan & Jaggi (2023) quantifies example learnability to prioritize those that may be most beneficial and Pouransari et al. (2025) uses variable sequence length and batch-size to improve long-context modeling. Wang et al. (2025) applies a similar approach for in-context learning. Other works are draw on *importance or loss-based sampling*, where training examples are weighted by estimated contribution to learning progress. At large scale, related ideas appear in corpus-level mixture design, such as the fixed data-mixture weights used to balance sources in GPT-3 (Brown et al., 2020), which adjust the proportions of different domains to achieve a desired training distribution. Some methods address corpus in a heterogeneous way through balanced or cluster-based selection, which preserves diversity by balancing common and rare examples to improve model training(). Recently, Shao et al. (2024) introduces *distribution-level curriculum learning* for LLM post-training, dynamically balancing exploration and exploitation across reward-conditioned datasets. While these approaches significantly reduce training cost or improve data efficiency, they generally assume offline access to the full corpus or slowly changing data distributions. They thus differ from our setting, which requires adaptive decisions on streaming data with a limited annotation budget.

## B    DATASETS AND LLM COLLECTIONS

More than 130 LLMs over 10 different datasets have been employed including different generative tasks. Except for LLMs of MT-Bench, which were evaluated using BERTScore (Zhang et al., 2020), the performance of the LLMs on the test sets of the other datasets was assessed using ROUGE-L (Lin, 2004).

| Dataset | No. of instances | No. of LLMs | LLM scores |
|---|---|---|---|
| Comprehensive–Arithmetic–Problems | 700 | 13 | 0.17 - 0.63 |
| Calculus Dataset (FDU) | 500 | 5 | 0.84 - 0.95 |
| LongForm–QA (Prompt) | 850 | 11 | 0.02 - 0.54 |
| Grammar Correction | 2000 | 5 | 0.76 - 0.84 |
| MedCalc | 750 | 9 | 0.12 - 0.41 |
| MT–Bench | 78 | 6 | 0.76 - 0.79 |
| SQuAD | 5000 | 7 | 0.47 - 0.93 |
| SQuAD v2 | 5000 | 5 | 0.39 - 0.77 |
| LongForm–QA (Canonical) | 600 | 54 | 0.16 - 0.66 |
| FinQA | 500 | 22 | 0.39 - 0.81 |

Table 2: Overview of datasets and LLM collections, including number of models and the scores achieved on the test set. The number of instances represents the stream length.

Figure 4 shows the distribution of LLM scores across datasets. Depending on the task, scores are computed with ROUGE-L (Lin, 2004) or BERTScore (Zhang et al., 2020). We evaluate a diverse collection of LLMs, yielding varied score distributions and enabling a comprehensive assessment. As reported in Table 2, the observed scores span 0.02–0.97 across models and datasets. Where available, we use benchmarks that release both LLM outputs and oracle annotations. In contrast, for the Calculus Dataset (FDU) (Zhang, 2025), Grammar Correction (Agentlans, 2024), Comprehensive Arithmetic Problems (Lee, 2024), SQuAD (Rajpurkar et al., 2016), and SQuAD v2 (Rajpurkar et al., 2018), only oracle annotations are provided; therefore we ran LLMs ourselves to generate the model outputs. As detailed in Section 4, this includes instruction-tuned LLMs (to reduce output noise and elicit agent-specific behavior), models fine-tuned on the corresponding datasets, and off-the-shelf

general LLMs. Our study is model-agnostic: we make no assumptions about LLM architectures or task-specific use cases and assess models in a manner consistent with realistic practitioner work-flows.

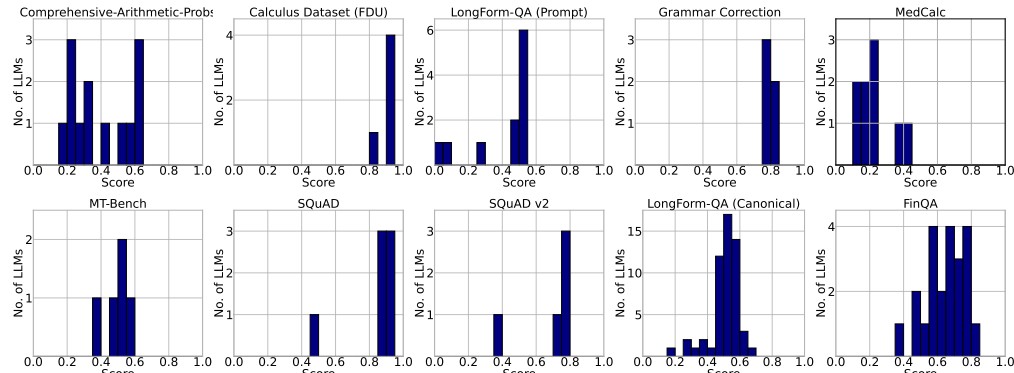

Figure 4: Score of LLM collections across the datasets.

## C    BASELINES

**Disagreement** Similar to the Random algorithm, the probability of querying the annotation at each time instance is fixed to $q_t = b/T$. However, we restrict our attention to instances where there is significant disagreement among experts. Specifically, cases in which the answers generated by the LLMs differ substantially from one another. We quantify disagreement at time instance $t$ as the variance—across experts—of each expert's average pairwise discrepancy from the others in the set of generated answers. Denote this statistic by $\Xi_t$. An instance is deemed to exhibit disagreement whenever $\Xi_t$ exceeds a small near-zero threshold $\delta$.

**Kullback–Leibler (KL)** We maintain a belief $\boldsymbol{p}_t$ over the $m$ models from past annotations (Hedge-style posterior). At each time instance $t$, we turn the mean of the pairwise losses among the answers generated by the LLMs into a normalized predictive loss distribution $\tilde{p}_t$ (via a softmax over the losses) and measure its mismatch from our belief with $D_t = \mathrm{KL}(\tilde{p}_t \,\|\, p_t)$ (Shlens, 2014). The query probability is set proportional to $D_t$, so we are more likely to annotate when the present loss pattern looks uncertain under our belief and less likely when LLMs agree. We sample a Bernoulli query with bias $D_t$ and spend from a fixed budget $b$ until exhaustion.

**Uncertainty** At each time instance $t$, we form a per–model average loss profile $p_t$ by averaging the current (unlabeled) pairwise prediction losses across models. We then compute an entropy–like uncertainty score $H_t = -\frac{1}{m}\sum_i p_{t,i} \log(p_{t,i})$ and set the query probability $q_t$ proportional to $H_t$. Thus, instances that bring higher uncertainty are queried more often. We sample a Bernoulli query with bias $H_t$ and spend from a fixed budget $b$ until exhaustion.

## D    EXTENDED RESULTS

This section presents additional results and comprehensive data from the experiments described in Section 4. Separately for each dataset and for each budget level, we conduct 500 independent realizations, ensuring robust and reliable results.

### D.1    REGRET

We evaluate the regret of ONLINE LLM PICKER and baselines at different budget levels where our method identifies the best LLM with high confidence. Table 3 reports the budget levels used for each dataset. Across datasets, ONLINE LLM PICKER attains consistently lower regret than the baselines. For each dataset, we compare ONLINE LLM PICKER against the strongest competing baseline at the same budget and report the reduction factor (best-baseline regret divided by ONLINE LLM PICKER regret). As shown in Table 3, ONLINE LLM PICKER achieves regret reductions of

$2.51\times$, $2.47\times$, $2.44\times$, and $1.97\times$ on Comprehensive–Arithmetic–Problems, SQuAD, SQuAD v2, and FinQA, respectively.

| Dataset | Budget level | Regret |
|---|---|---|
| Comprehensive–Arithmetic–Problems | 120 | $\downarrow$ **2.51** $\times$ |
| Calculus Dataset (FDU) | 65 | $\downarrow$ **1.93** $\times$ |
| LongForm–QA (Prompt) | 450 | $\downarrow$ **1.17** $\times$ |
| Grammar Correction | 1200 | $\downarrow$ **1.01** $\times$ |
| MedCalc | 400 | $\downarrow$ **1.54** $\times$ |
| MT–Bench | 50 | $\downarrow$ **1.19** $\times$ |
| SQuAD | 450 | $\downarrow$ **2.47** $\times$ |
| SQuAD v2 | 800 | $\downarrow$ **2.44** $\times$ |
| LongForm–QA (Canonical) | 300 | $\downarrow$ **1.58** $\times$ |
| FinQA | 200 | $\downarrow$ **1.97** $\times$ |

Table 3: Budget levels and regret analysis.

| Dataset | Online LLM Picker | Kullback–Leibler | Uncertainty | Disagreement | Random |
|---|---|---|---|---|---|
| Comprehensive-Arithmetic-Problems | **3.67** | 35.39 | 10.09 | 9.89 | 9.22 |
| Calculus Dataset | **0.82** | 7.75 | 1.58 | 3.62 | 3.01 |
| LongForm-QA (Prompt) | **8.45** | 24.06 | 10.31 | 9.88 | 9.97 |
| Grammar Correction | **2.25** | 5.07 | 2.27 | 2.52 | 2.31 |
| MedCalc | **4.57** | 29.18 | 9.67 | 7.03 | 7.07 |
| MT-Bench | **1.72** | 2.71 | 2.06 | 2.29 | 2.14 |
| SQuAD | **8.78** | 79.70 | 21.67 | 28.93 | 29.58 |
| SQuAD v2 | **5.98** | 48.45 | 14.58 | 18.83 | 19.31 |
| LongForm–QA (Canonical) | **5.39** | 26.98 | 9.63 | 8.88 | 8.53 |
| FinQA | **3.10** | 21.04 | 14.83 | 6.10 | 6.44 |

Table 4: Regret of ONLINE LLM PICKER and baselines. Bold denotes the best value, underlining denotes the second-best. The regret of ONLINE LLM PICKER is consistently lower with respect to baselines.

## D.2 IDENTIFICATION PROBABILITY

An overview of the percentage of budget needed for each method to reach the 100% identification probability for the first time can be found in Table 5. Note that the percentage of budget for each method is now computed with respect to the total length of the stream. ONLINE LLM PICKER consistently requires less annotations compared to baselines to reach the maximum identification probability, showing its effectiveness in properly selecting the most informative annotations to improve the selection strategy.

| Dataset | Online LLM Picker | Kullback–Leibler | Uncertainty | Disagreement | Random |
|---|---|---|---|---|---|
| Comprehensive-Arithmetic-Problems | **28.26**% | 97.10% | 100.00% | 91.30% | 95.65% |
| Calculus Dataset | **15.00**% | 80.00% | 38.00% | 80.00% | 90.00% |
| LF-QA-Prompt | **78.82**% | 100.00% | 99.41% | 100.00% | 99.41% |
| Grammar Correction | **74.00**% | 98.00% | 74.50% | 100.00% | 98.00% |
| MedCalc | **78.00**% | 100.00% | 98.00% | 100.00% | 100.00% |
| MT-Bench | **71.79**% | 96.15% | 98.71% | 92.30% | 93.58% |
| SQuAD | **13.00**% | 90.00% | 29.00% | 80.00% | 58.00% |
| SQuAD v2 | **17.00**% | 100.00% | 60.00% | 90.00% | 80.00% |
| LongForm–QA (Canonical) | **84.16**% | 100.00% | 98.33% | 96.67% | 99.17% |
| FinQA | **47.00**% | 88.00% | 100.00% | 84.00% | 72.00% |

Table 5: Percentage of annotations (w.r.t. total stream length) required by ONLINE LLM PICKER and baselines to reach 100% identification probability. Bold denotes the best value, underlining denotes the second-best.

### D.3 Robustness Analysis

We compute the 95-th percentile accuracy gap at budget needed by ONLINE LLM PICKER to reach certain identification probability levels. Specifically, we focus on 70%, 80%, 90% and 100% identification probabilities. If the exact desired identification probability value is unavailable, the next higher closest value is used. Table 6 shows the results. Best values are in bold, second-best values are underlined. In 36 cases out of 40, ONLINE LLM PICKER achieves the lowest 95-th percentile accuracy gap. However, even for the four remaining cases, ONLINE LLM PICKER performs competitively, achieving the second lowest accuracy gap.

| Dataset
Identification probability | Online LLM Picker
(70% / 80% / 90% / 100%) | Kullback–Leibler
(70% / 80% / 90% / 100%) | Uncertainty
(70% / 80% / 90% / 100%) | Disagreement
(70% / 80% / 90% / 100%) | Random
(70% / 80% / 90% / 100%) |
|---|---|---|---|---|---|
| Comprehensive-Arithmetic-Problems | 1.81 / **1.33** / **0.86** / **0.00** | 3.71 / 2.67 / 2.57 / 2.19 | 2.10 / 1.71 / 1.71 / 1.43 | **1.62** / 1.62 / 1.33 / 1.43 | 1.81 / 1.71 / 1.52 / 1.24 |
| Calculus Dataset | 1.12 / **0.91** / **0.66** / **0.00** | 3.11 / 3.03 / 2.85 / 2.76 | **1.02** / 0.95 / 0.87 / **0.00** | 2.92 / 2.65 / 1.22 / 0.88 | 2.81 / 2.46 / 1.10 / 0.91 |
| LongForm–QA (Prompt) | **1.45** / **1.23** / **0.63** / **0.00** | 2.89 / 1.82 / 1.68 / 0.98 | 1.72 / 1.43 / 1.35 / 0.85 | 1.53 / 1.54 / 1.26 / 0.69 | 1.59 / 1.37 / 1.30 / 0.65 |
| Grammar Correction | **0.11** / **0.11** / **0.11** / **0.00** | **0.11** / **0.11** / **0.11** / 0.11 | **0.11** / **0.11** / **0.11** / **0.00** | **0.11** / **0.11** / **0.11** / 0.11 | **0.11** / **0.11** / **0.11** / 0.11 |
| MedCalc | 2.13 / **1.84** / **0.99** / **0.00** | 4.40 / 2.62 / 2.24 / 0.48 | 2.55 / 2.45 / 2.44 / 1.59 | 2.37 / 2.19 / 1.99 / 0.89 | **2.10** / 1.99 / 1.79 / **0.00** |
| MT-Bench | **6.61** / **3.86** / **3.28** / **0.00** | 7.59 / 7.50 / 6.73 / 3.79 | 7.03 / 4.78 / 3.92 / 3.71 | 6.93 / 6.42 / 3.83 / 2.82 | 6.99 / 4.61 / 3.86 / 3.01 |
| SQuAD | 1.69 / **1.33** / **1.12** / **0.00** | 2.05 / 1.95 / 2.15 / 1.72 | **1.54** / 1.48 / 1.37 / 0.57 | 1.74 / 1.68 / 1.61 / 1.24 | 1.76 / 1.66 / 1.61 / 1.24 |
| SQuAD v2 | **1.30** / **0.92** / **0.65** / **0.00** | 1.77 / 1.58 / 1.44 / 1.06 | 1.31 / 1.21 / 0.94 / 0.78 | 1.49 / 1.28 / 1.18 / 0.92 | 1.39 / 1.29 / 1.10 / 0.85 |
| LongForm–QA (Canonical) | **2.91** / **2.47** / **1.41** / **0.00** | 7.78 / 7.46 / 4.22 / **0.00** | 4.11 / 3.42 / 2.59 / 0.43 | 3.21 / 2.79 / 2.01 / **0.00** | 3.51 / **3.46** / 1.96 / **0.00** |
| FinQA | **3.56** / **2.92** / **1.99** / **0.00** | 6.72 / 4.26 / 5.21 / 2.74 | 3.67 / 3.67 / 3.83 / 3.96 | 3.67 / 3.33 / 2.74 / **0.00** | **3.56** / 3.29 / 2.98 / **0.00** |

Table 6: Robustness analysis: 95-th percentile accuracy gap at the budget needed for ONLINE LLM PICKER to reach identification probabilities of 70%, 80%, 90%, and 100%. Bold denotes the best value, underlining denotes the second-best.

### D.4 Annotation Efficiency

For completeness, we show the extended plots related to the results of annotation efficiency described in Table 1.

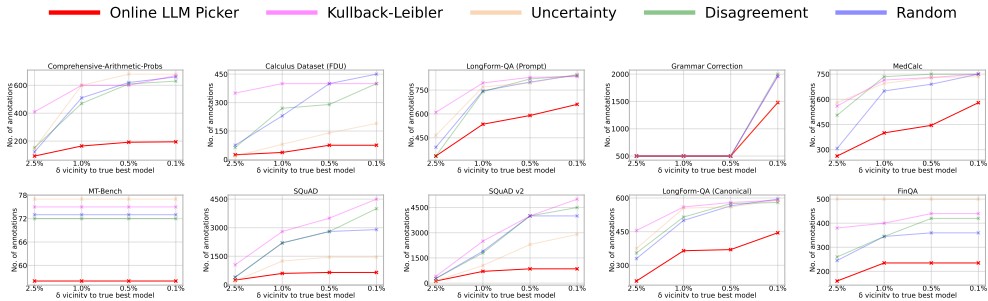

Figure 5: Identification probabilities of ONLINE LLM PICKER and baselines across different annotation budgets.

## E Additional Details

Let $\hat{L}_{t-1,i}$ be the cumulative loss estimate and $\eta_t$ the adaptive learning rate. The posterior distribution $\mathbf{p}_t = [p_{t,i}]_{i \in \mathcal{M}}$ is updated as:

$$p_{t,i} = \frac{\exp\left\{-\eta_t \hat{L}_{t-1,i}\right\}}{\sum_{j \in \mathcal{M}} \exp\left\{-\eta_t \hat{L}_{t-1,j}\right\}} \tag{11}$$

