# OpenReview forum: "Which LLM to pick? Online Active Model Selection for Large Language Models"
_ICLR.cc/2026/Conference — Submitted to ICLR 2026_

### Official Review · Reviewer_hBS5 · 2025-10-28

**Soundness:** 3
**Presentation:** 1
**Contribution:** 2
**Rating:** 2
**Confidence:** 4

**Summary:**

Picking which LLM to serve in an online setting is an important problem inspired by real-world use cases. LLM Picker attempts to solve this problem with guarantee by using an algorithm based on the EW algorithm with a modified adaptive learning rate rule. Experiments to demonstrate the advantage over baseline algorithms are provided.

**Strengths:**

- Well-defined metrics for the problem.
- Good performance on several benchmarks with a variety of models.

**Weaknesses:**

- The writing could be much better. Sec 3.2 reads like a laundry list of the things the author did without a clear story and explanation for the various designs of the algorithm.
- Unclear notation: $p_t$ is introduced on line 133 before definition.
- Section 3.2: the authors should formally state why the EW algorithm is chosen to minimize regrets.
- There is actually a related work using the EW algorithm for the same problem [1].
[1] Feng, Y., Khare, A., Nguyen, N., & Sengupta, S. B. Boss LLM: Adaptation via No-Regret Learning. In ICLR Scaling Self-Improving Foundation Models without Human Supervision Workshop.

**Questions:**

- In Sec 3.1, what are the verifiable advantage of the new adaptive learning rate rule vs pure EW rule?
- In Eq 10, what are the final contributions of the different design choices to the final performance? Is there an ablation study?

---

> ### Author Response · Authors · 2025-11-26
> **Official Comment by Authors**
>
> Thank you for your review, we try to address your concerns and questions below.
>
> > The writing could be much better. Sec 3.2 reads like a laundry list of the things the author did without a clear story and explanation for the various designs of the algorithm.
>
> Thank you for your feedback. We now provide an explanation of the reason why we decided to write Section 3.2 using this format. We structured Section 3.2 in this way to provide a sequential walkthrough of our algorithm by incrementally introducing relevant design choices. Each passage of this section presents additional details useful to the explanation of our method and reasoning, rigorously justifying our design choices and implementations.
>
> In particular, our idea was to first provide a general overview of how our method works with respect to the problem setting, as depicted at the beginning of Section 3.2. We first explain that we decide whether to query or not an annotation $y_t$ for a given prompt $x_t$ based on the query probability $q_t$, which is a common practice in online model selection. Subsequently, we introduce our posterior distribution over models $p_t$, which updates our belief over the set of LLMs, giving more importance to those models that performed well in the past. The distribution $p_t$ is updated using the Exponential Weights (EW) algorithm with learning rate $\eta_t$ (Subsection 3.2.1), which depends on the exponential moving average of the proxy of the variance of the examples observed up to t. Finally, Subsection 3.2.2 describes the construction of $q_t$, which is based on a balanced combination of the entropy of the posterior distributions over models and the hypothetical loss variance.
>
> To improve the narrative flow, we will revise Section 3.2 by adding a concise high-level overview and clarifying design choices that may be unclear, making the presentation of the workflow more coherent to the reader. Importantly, the method and its design choices are principled and rigorously grounded, and these revisions do not affect the validity of our approach.
>
> > w_t not defined before line 133 and EW choice
>
> Thanks for pointing it out.
> $p_t$​ denotes the posterior distribution over candidate models at round $t$, which evolves according to the update rule described in Equation 3. We agree that the notation for $p_t$ introduced in line 156 may be too late relative to its first use in line 133. In the camera ready version, we will clarify the meaning of $p_t$ in line 133. This correction improves clarity without altering any technical content.
> We use Exponential Weights because it is the standard and minimax-optimal update for online learning on the simplex. EW preserves non-negativity and normalization without requiring projections, and it yields the classical $O(\sqrt{T\log⁡N})$ regret bound. This makes it a natural and widely accepted choice in our setting.
>
> > There is actually a related work using the EW algorithm for the same problem [1]. [1] Feng, Y., Khare, A., Nguyen, N., & Sengupta, S. B. Boss LLM: Adaptation via No-Regret Learning. In ICLR Scaling Self-Improving Foundation Models without Human Supervision Workshop.
>
> Thank you for pointing out this additional related work that employs the Exponential Weights (EW) algorithm, an approach that has also been explored in online active model selection settings, such as Karimi et al. (2021), Online Active Model Selection for Pre-trained Classifiers, and Liu et al. (2024), Contextual Active Model Selection. However, we respectfully disagree with the assertion that Boss LLM addresses the same problem as our work. Boss LLM is fundamentally a mixture-of-experts (MoE) framework aimed at dynamically selecting or weighting multiple LLMs to optimize prediction quality for each prompt.
>
> In contrast, Online LLM Picker tackles a different challenge: online active LLM selection, where the goal is to strategically query a small number of highly informative annotations to efficiently identify the best LLM for a stream of prompts. MoE methods focus on per-prompt expert selection at inference time, whereas our method focuses on annotation efficiency and the rapid identification of the best model. These two problem formulations and objectives are therefore distinct and not directly comparable.
>
> > In Sec 3.1, what are the verifiable advantages of the new adaptive learning rate rule vs pure EW rule?
>
> The new adaptive learning rate is based on the proxy of the variances of those annotations that are observed. We aim to query only those annotations that may be informative for the LLM selection strategy. The learning rate is dynamically updated according to the variances of the observed losses in the effective windows, which provides a form of memory to the system. This enables the updates in the posterior to be stronger or leaner based on previously observed examples, making the
> selection process more responsive and efficient compared to the pure EW rule.

---

> > ### Author Response · Authors · 2025-11-26
> > **Official Comment by Authors**
> >
> > > In Eq 10, what are the final contributions of the different design choices to the final performance? Is there an ablation study?
> >
> > Looking at the ablation study as a whole, we can frame the results in a way that highlights both the strengths of the method and the rationale behind each design choice. Let’s take a look at the Comprehensive Arithmetic Problems dataset. First, all versions of the algorithm consistently outperform the baseline methods, which already indicates that the overall structure of the approach is sound and effective.
> >
> > When we break things down further, the largest improvements clearly come from combining variance and entropy. This component has the strongest impact across the board, so emphasizing its contribution feels well-justified.
> >
> > Regarding the adaptive learning rate, the full Online LLM Picker reaches 100% identification probability at around 196 annotations, whereas the version without adaptive LR needs about 202. The difference is small, but since we’re essentially comparing variants of our own method, even slight gains matter in an ablation study. The important point is that the adaptive LR generally leads to improvements. Eventually, we want to remark that the adaptive learning rate depends on the proxy of the variance of those annotations that are observed, thus performance gains also depend on the data distribution under analysis.
> >
> > When evaluating generative language models, a wide range of metrics can be used. Traditional approaches rely on statistical measures such as BLEU and ROUGE, while more recent methods include sentence-embedding–based metrics and even LLM-as-a-judge evaluations. However, these metrics can become unreliable when the data comes from highly specialized domains. In such cases, general-purpose embedding models or LLM-based judges may fail to capture true semantic similarity, resulting in noisy or misleading scores. In practice, the choice of an appropriate metric is highly context-dependent and closely tied to the underlying data distribution. For example, in our experiments with medical datasets, several open source embedding models produced nearly constant similarity scores across sentences, effectively reducing the algorithms’ behavior to randomness. Identifying robust, domain-appropriate evaluation metrics is an active area of research for many generative tasks, but a detailed investigation of this topic lies beyond the scope of our current work.
> >
> > A section including the ablation study will be added in the appendix for the camera-ready version.

---

### Official Review · Reviewer_nkAt · 2025-10-31

**Soundness:** 3
**Presentation:** 2
**Contribution:** 2
**Rating:** 2
**Confidence:** 3

**Summary:**

This paper proposes an online LLM model router that selects which model is best for a certain input.

**Strengths:**

The paper compares their online LLM picker against several baselines and strategies. They show that their approach is best among all in a set of single-turn standard benchmarks.

The models selected for the experiment include both open-source and proprietary LLMs (see lines 345-358). This makes the experimental setup thorough

**Weaknesses:**

The paper claims that this is the first framework for LLM routing,  there is work on this area e.g., https://arxiv.org/abs/2309.15789, https://arxiv.org/abs/2402.05859, https://proceedings.neurips.cc/paper_files/paper/2024/file/7a641b8ec86162fc875fb9f6456a542f-Paper-Conference.pdf. The paper does not cite these and others.

The paper only tackles a collection of public standard benchmarks which makes the scope and impact of the experiments quite limited.

Given that the paper claims that this is an online routing, it would be great to showcase this in the context of multi-turn trajectories (this paper could be a good start https://arxiv.org/pdf/2505.06120). Maybe each turn can be sent to a different LLM.

**Questions:**

How do you account for potential contamination of the routing (i.e., the model used to route knows about the benchmark) and the underlying models have been trained on the benchmarks. It would be great to show datasets that are unique for this task.

How would this work in a conversational setting with multiple trajectories (see e.g., https://arxiv.org/pdf/2505.06120) ?

How would this work in an agent environment with react framework, would it be possible to also pick LLMs in that context?

I saw this comment in line 354: Prompt engineering is also used to improve performance on our benchmark datasets by reducing unnecessary verbosity and avoiding irrelevant explanations in the output.
could you elaborate? does this make the results not comparable with other methods using teh same benchmarks?

In Figure 2, what are the results of always taking the top performing model?

---

> ### Author Response · Authors · 2025-11-26
> **Official Comment by Authors**
>
> Thank you for your review. First, we would like to clarify that we do **not claim** to be the first to propose an LLM routing approach; our contribution is instead the first online model selection strategy for LLMs under a limited annotation budget. **This setting is fundamentally different from LLM routing and addresses a separate research problem**. We will now highlight for you the main differences with respect to our setting:
> - Routing methods typically aim to decide which model should answer each individual query, often using heuristics, embeddings, or simple decision rules.
> - Crucially, routing does not rely on ground-truth annotations at all, nor does it operate under an annotation budget (annotations are simply not part of the problem formulation). In contrast, our framework assumes that ground-truth annotations exist but are costly, and therefore must be queried selectively.
> - The objective in our setting is to identify the best LLM overall in a streaming environment while respecting a strict annotation budget.
> - Our method explicitly focuses on determining which annotations are most informative to query based on the incoming prompts, in order to make the best possible use of a limited subset of them.
> - Routing methods do not perform any form of selective annotation acquisition, since they do not use annotations in the first place.
> - Our setting requires active querying, online learning, and posterior updates based on careful use of annotations, not per-query dispatching or model routing. For these reasons, existing LLM routing techniques do not apply to our problem formulation, nor do they address the central challenge we target: reliable model selection under constrained annotation resources.
> - Our approach is specifically designed for this annotation-efficient setting.

---

> > ### Comment · Reviewer_nkAt · 2025-11-26
> >
> > I understand that your work is on online active model selection, but you are still routing queries (or subqueries) to different models. I would recommend including what you describe in your response as part of the context by citing those or other papers related to LLM routing. I don't think you should ignore highly relevant and related work, even if  you are addressing a novel problem within the same realm of topics. Mainly because that novel problem is highly related to the routing problem discussed.
> >
> > re: Crucially, routing does not rely on ground-truth annotations at all.
> >
> > https://arxiv.org/abs/2412.04167 uses benchmark evaluations to train routers, while https://arxiv.org/pdf/2311.08692 employs reward-guided mechanisms to train routers and route inputs effectively.

---

> > > ### Author Response · Authors · 2025-11-27
> > > **Official Comment by Authors**
> > >
> > > Thank you for the follow-up. We have now included citations to LLM routing work and added a dedicated paragraph that explains how our setting differs, so that the paper is more clearly positioned within that literature.
> > >
> > > To clarify the technical distinction: in our framework, each prompt $x_t$ is evaluated by all candidate LLMs. Based on the resulting outputs, we then decide whether to query the ground-truth annotation for that prompt under a strict annotation budget. Our objective is to identify a single best LLM for the stream, not to decide which model should answer a specific user query at inference time.
> > >
> > > By contrast, routing methods (including the ones you cite) are designed to train a router that dispatches each incoming query to one model (or a subset of models). Even when they leverage benchmark evaluations or reward signals, this supervision is used to train the router itself, not to perform selective, budgeted annotation of the stream. In particular, these methods do not implement selective annotation acquisition as a core algorithmic component, whereas our method is built precisely around which labels to acquire under a tight budget.
> > >
> > > For these reasons, we view routing approaches as conceptually related but ultimately complementary, rather than directly comparable baselines for our annotation-efficient online model selection problem. We will make this distinction explicit in the revised related-work section and throughout the paper to avoid any confusion about the problem formulation.
> > >
> > > We hope that these clarifications and the corresponding revisions in the paper address your concerns and that you might consider revisiting your overall assessment in light of them.

---

### Official Review · Reviewer_4HJd · 2025-11-01

**Soundness:** 2
**Presentation:** 3
**Contribution:** 3
**Rating:** 4
**Confidence:** 4

**Summary:**

This paper tackle the problem of active model selection for LLMs in a online streaming settings. It proposes a new framework named online LLM picker, which picks next prompt for oracle annotation based on the information gain to identify the best LLM. The proposed approach combines an Exponential Weights algorithm with adaptive learning rate based on loss variance, together with a query probability that balances hypothetical variance across model responses with posterior entropy. The paper performs experiments on 10 datasets and 130+ LLMs show up to 71.6% annotation savings and 2.51x regret reduction compared to selected baselines.

**Strengths:**

The paper addresses the problem of active model selection in an online streaming fashion which is well formulated and resonate well with real world applications, especially given the proliferation of available LLMs and the cost of annotations. The proposed framework is model agnostic (no assumptions about the LLM architecture or require log likelihood) and it works with both black and white box LLMs. This ensures the general applicability of the approach. The paper performs experiments on 10 datasets among diverse tasks over 130+ LLMs, the results show quite strong and consistent improvements over baselines in terms of the annotation efficiency (up to 71.6% savings) and identification probabilities, which proves the great value of the proposed approach in the real world settings.

**Weaknesses:**

1. Missing baselines in the comparison. Although the authors claim they are the first paper deal with active model selection in an online streaming fashion, I would still argue the baselines are still quite simple and it should have compared with other active model selection methods. For example, the paper cites Ashury-Tahan et al. 2024 (any many others) in the related work, these method can be easily adapted to the online streaming fashion by applying sliding window to convert from pool based method to the streaming based method. Comparison with these SOTA methods will certainly informative.

2. Missing results analysis and ablation study. The paper introduces some variable components in the framework, such as the adaptive learning rate (vs. fixed learning rate), variance only vs. variance + entropy and various of similarity metrics for loss calculation. What is the impact for each of them? How sensitive will the final results be due to the change of these choices? These questions are not discussed in the paper.

3. The experimentation setup for the online streaming simulation is limited. Part of the novelty for the paper comes from the online streaming model selection, but I don't think we are simulating some unique patterns potentially from the stream. For example, what if the underlying tasks from the stream is too diverse? How will that impact the performance? What if the stream has significant distribution shift over time? How will the method generalizes to these distribution shift? I would recommend to design some experiments to show specific characteristics w.r.t the streaming vs. a fixed pool.

**Questions:**

1. We are using pretty "old" similarity based metrics such as ROUGE/BERTScore etc. in the paper. These metrics are proven to less correlated with human performance. What is the reason we select these metrics? For example, LLM based metrics correlated much better with human, what is the reason we didn't select these better metrics? Have we tried these metrics and how that impact the model performance comparison?

2. Do we have any theoretical analysis to show the bound of the online LLM picker, for example, on the regret? I think the work can get strengthened even if we provide some simple analysis there.

3. Have we compared the method with other pool based active model selection method? How does it compare with these SOTA models? I think some analysis there will be very helpful for the community to understand the difference and the value provided in the paper.

---

> ### Author Response · Authors · 2025-11-26
> **Official Comment by Authors**
>
> Thank you for your thoughtful review and comments. We appreciate your recognition of the practical value of our proposed approach in real-world settings. We address your concerns and questions below.
>
> > Missing baselines in the comparison. Although the authors claim they are the first paper deal with active model selection in an online streaming fashion, I would still argue the baselines are still quite simple and it should have compared with other active model selection methods. For example, the paper cites Ashury-Tahan et al. 2024 (any many others) in the related work, these method can be easily adapted to the online streaming fashion by applying sliding window to convert from pool based method to the streaming based method. Comparison with these SOTA methods will certainly informative.
>
> We agree that prior work on active model selection, including Ashury-Tahan et al. (2024), is closely related. However, there are two main reasons we do not include these methods as baselines in our experiments.
>
> First, most existing methods are designed for classification and assume access to per-class probabilities,  or measures such as Shannon entropy. In our setting, we work with open-ended text generation, using model-generated quality scores such as ROUGE or BERTScore, where outputs are unconstrained and no discrete class set is available. These methods are therefore not well-defined for our setting, and require substantial redesign rather than a straightforward adaptation.
>
> Second, these works, such as Ashury-Tahan et al. (2024), operate in a pool-based setting, where the algorithm can inspect the outputs of all candidate models on a large set of examples before deciding what to annotate. Our method, in contrast, is explicitly stream-based: it must decide irrevocably for each incoming instance, without observing future examples or performing cross-example comparisons. A naive sliding-window variant of a pool-based method would still assume significantly more information than is available in our setting and thus would not constitute a fair or principled baseline. We elaborate on this pool-vs-stream distinction in our response to the subsequent question.
>
> If the reviewer has specific suggestions that can be adapted to our generative, streaming setting, we would be happy to consider and incorporate them where feasible.

---

> > ### Author Response · Authors · 2025-11-26
> > **Official Comment by Authors**
> >
> > > Missing results analysis and ablation study. The paper introduces some variable components in the framework, such as the adaptive learning rate (vs. fixed learning rate), variance only vs. variance + entropy and various of similarity metrics for loss calculation. What is the impact for each of them? How sensitive will the final results be due to the change of these choices? These questions are not discussed in the paper.
> >
> > Looking at the ablation study as a whole, we can frame the results in a way that highlights both the strengths of the method and the rationale behind each design choice. Let’s take a look at the Comprehensive Arithmetic Problems dataset. First, all versions of the algorithm consistently outperform the baseline methods, which already indicates that the overall structure of the approach is sound and effective.
> >
> > When we break things down further, the largest improvements clearly come from combining variance and entropy. This component has the strongest impact across the board, so emphasizing its contribution feels well-justified.
> >
> > Regarding the adaptive learning rate, the full Online LLM Picker reaches 100% identification probability at around 196 annotations, whereas the version without adaptive LR needs about 202. The difference is small, but since we’re essentially comparing variants of our own method, even slight gains matter in an ablation study. The important point is that the adaptive LR generally leads to improvements. Eventually, we want to remark that the adaptive learning rate depends on the proxy of the variance of those annotations that are observed, thus performance gains also depend on the data distribution under analysis.
> >
> > When evaluating generative language models, a wide range of metrics can be used. Traditional approaches rely on statistical measures such as BLEU and ROUGE, while more recent methods include sentence-embedding–based metrics and even LLM-as-a-judge evaluations. However, these metrics can become unreliable when the data comes from highly specialized domains. In such cases, general-purpose embedding models or LLM-based judges may fail to capture true semantic similarity, resulting in noisy or misleading scores. In practice, the choice of an appropriate metric is highly context-dependent and closely tied to the underlying data distribution. For example, in our experiments with medical datasets, several open source embedding models produced nearly constant similarity scores across sentences, effectively reducing the algorithms’ behavior to randomness. Identifying robust, domain-appropriate evaluation metrics is an active area of research for many generative tasks, but a detailed investigation of this topic lies beyond the scope of our current work.
> >
> > A section including the ablation study will be added in the appendix for the camera-ready version.

---

> ### Author Response · Authors · 2025-11-26
> **Official Comment by Authors**
>
> > The experimentation setup for the online streaming simulation is limited. Part of the novelty for the paper comes from the online streaming model selection, but I don't think we are simulating some unique patterns potentially from the stream. For example, what if the underlying tasks from the stream is too diverse? How will that impact the performance? What if the stream has significant distribution shift over time? How will the method generalizes to these distribution shift? I would recommend to design some experiments to show specific characteristics w.r.t the streaming vs. a fixed pool.
>
> Some works in the classification setting have indeed considered datasets involving substantial distribution shifts, such as the DRIFT dataset. However, to the best of our knowledge, no public benchmarks exist that capture distribution shift in the generative setting. One might argue that we could construct such a dataset ourselves, but doing so reliably is substantially more challenging than in classification. In classification, labels are discrete and the associated losses are well-defined, making it relatively straightforward to engineer controlled shifts. In contrast, generative outputs are long, open-ended texts, and the evaluation metrics are continuous and often noisy (e.g., ROUGE, BERTScore). As a result, creating a principled and reproducible distribution shift for generative model outputs is non-trivial and risks introducing artifacts rather than meaningful shifts in the data distribution.
>
> Regarding the use of different underlying tasks, we included MT-Bench, which spans a diverse set of domains (such as writing, reasoning, math, coding, and role-playing). This diversity provides a broad evaluation across multiple task types within a single benchmark, helping validate that our method is not tied to a specific generative task distribution.
>
> > We are using pretty "old" similarity based metrics such as ROUGE/BERTScore etc. in the paper. These metrics are proven to less correlated with human performance. What is the reason we select these metrics? For example, LLM based metrics correlated much better with human, what is the reason we didn't select these better metrics? Have we tried these metrics and how that impact the model performance comparison?
>
> We acknowledge that ROUGE and BERTScore are not perfect proxies for human judgment; however, they remain widely used and not obsolete in current evaluation pipelines for generative models (e.g. (1) and (2)) continue to rely on them due to their stability, simplicity, and reproducibility. Moreover, several recent studies show that LLM-based metrics, while often better correlated with human evaluation, can also introduce significant variability, sensitivity to prompt phrasing, and model bias. We experimented with sentence-embedding–based similarity metrics, but found them to be considerably noisier and less stable across streaming batches. In practice, when we computed pairwise similarities between model answers, the embedding scores tended to cluster tightly, with very little spread, even in cases where the answers were meaningfully different. This suggests that the sentence-embedding model wasn’t reliably distinguishing between genuinely similar and dissimilar responses. In addition, LLM-based evaluators are substantially more expensive, both computationally and financially, and would make large-scale online evaluation impractical in our setting. For these reasons, we opted for ROUGE and BERTScore in this work, but we agree that evaluating our framework under more advanced LLM-based metrics is a valuable direction for future work.
>
> References:
> 1) SummaReranker: A Multi‑Task Mixture‑of‑Experts Re‑ranking Framework for Abstractive Summarization (Ravaut, Joty & Chen, 2022)
>
> 2) CUED at ProbSum 2023: Hierarchical Ensemble of Summarization Models (Manakul et al., 2023)
>
> > Do we have any theoretical analysis to show the bound of the online LLM picker, for example, on the regret? I think the work can get strengthened even if we provide some simple analysis there.
>
> In this work, we introduce the novel problem of online selection of pretrained LLMs and a principled and practical solution to address that by demonstrating its practical performance. A more extensive theoretical analysis is an extension of this work that we intend to pursue next.

---

> > ### Author Response · Authors · 2025-11-26
> > **Official Comment by Authors**
> >
> > > Have we compared the method with other pool based active model selection method? How does it compare with these SOTA models? I think some analysis there will be very helpful for the community to understand the difference and the value provided in the paper.
> >
> >  It is not a common practice in literature to compare stream-based methods against pool-based ones. That is because the learner inherently has more information in the pool-based setting as it can observe the generation of every model on every data instance. As such, the comparison would not favor the online methods.
> >
> > To open this up, pool-based methods assume access to the entire dataset onset and typically rely on comparing multiple candidate prompts and their corresponding outputs before selecting an annotation. In contrast, in the stream-based setting we consider, the algorithm must decide immediately and irreversibly whether to query an annotation for each incoming example, and no cross-example comparisons of alternative model outputs are available. For these reasons, simply applying a sliding-window variant of existing pool-based methods would not yield a meaningful or valid comparison. Ashury-Tahan et al., 2024 assumes that prompts are available from the offset and compares two candidate models over a certain batch to declare the winner, which substantially differs from the problem setting we are considering in our work.

---

### Official Review · Reviewer_NCH6 · 2025-11-02

**Soundness:** 3
**Presentation:** 3
**Contribution:** 3
**Rating:** 6
**Confidence:** 3

**Summary:**

The paper introduces ONLINE LLM PICKER, a framework for active model selection among multiple large language models (LLMs) in online settings where data arrive sequentially and annotation budgets are limited.
At each time step, the algorithm decides whether to query an oracle for the reference output, updates model posteriors using exponential-weights (Hedge) with an adaptive learning rate based on variance, inspired by AdaHedge ( Rooij et al., 2013 ), and uses a variance-entropy criterion to determine which prompts are most informative to label.
Compared to the current state-of-the-art active model selection methods, the proposed method follow a streamlined approach without relying on a given pool of labeled data.
Earlier methods rely on discrete class labels, making informativeness easy to define via label entropy. In contrast, this paper defines a variance-based informativeness measure over generated text, approximating the space of possible references using pairwise model outputs.
That allows the approach to be applied to open ended generative tasks like question answering, grammar correction, and arithmetic reasoning settings where prior active selection methods don't apply
The author conducts experiments on several generative tasks.
Across ten generative datasets, the method reportedly reduces annotation cost by up to 70.69 \% and reduces the regret compared with baselines such as Random, Disagreement, KL-divergence, and Uncertainty sampling significantly.

**Strengths:**

1. Prior active-selection work assumes pool-based settings or classification tasks; this is the first explicit treatment of streaming LLM selection with generation-style outputs.

2. The combination of Hedge-style posterior updates, adaptive learning rate $\eta_t$, and a joint variance-entropy query rule is mathematically clear and connects online learning with Bayesian experimental-design intuition.

3.	Comprehensive empirical validation.
Experiments span ten diverse generative datasets and more than 130 models. Metrics include regret, identification probability, and annotation efficiency, showing consistent gains (up to 2.5 $\times$ lower regret and $\approx$ 70 \% fewer annotations).

**Weaknesses:**

1. Although inspired by AdaHedge, the paper stops short of giving formal regret bounds for stochastic or adversarial streams. The adaptive $\eta_t$ and variance entropy querying are heuristic extensions without proofs of convergence or sample complexity guarantees.

2. The author uses several baselines (Random, Disagreement, KL-divergence, Uncertainty sampling) but does not give some details on their implementation. For example, how is uncertainty sampling defined for generative outputs? More clarity on baseline implementations would help assess the empirical gains.

3. Each query step requires evaluating $m \times m$ pairwise distances to estimate hypothetical variance (Eq. 8), which could be quadratic in the number of models. What is the computational overhead compared to baselines?

**Questions:**

Same as the weaknesses.

**Details Of Ethics Concerns:**

No ethics concerns.

---

> ### Author Response · Authors · 2025-11-26
> **Official Comment by Authors**
>
> Thank you for your thorough review and positive assessment of our work. We particularly appreciate your recognition of our paper's novelty and comprehensive empirical validation. Let us address the key weaknesses you raised.
>
> > Although inspired by AdaHedge, the paper stops short of giving formal regret bounds for stochastic or adversarial streams. The adaptive  and variance entropy querying are heuristic extensions without proofs of convergence or sample complexity guarantees.
>
> While additional convergence and sample-complexity guarantees would further develop the theoretical side of this work, in this work we introduce the novel problem of online selection of pretrained LLMs and a principled and practical solution to address that. A more extensive theoretical analysis is an extension of this work that we intend to pursue next.
>
> > The author uses several baselines (Random, Disagreement, KL-divergence, Uncertainty sampling) but does not give some details on their implementation. For example, how is uncertainty sampling defined for generative outputs? More clarity on baseline implementations would help assess the empirical gains.
>
> We describe our adaptations of these methods in detail primarily in the Appendix. To clarify the understanding from the main body of the paper, we summarize the key points below.
>
> The Uncertainty method queries when overall ambiguity about model performance is high. We form a per-model loss profile $p_t$​ from current models’ pairwise losses and compute an entropy-like uncertainty score $H_t$. Thus, we set the query probability $q_t$ proportional to $H_t$ and sample queries until the annotation budget runs out.
>
> The Disagreement method queries annotations only when the LLMs substantially diverge in their predictions. We measure this by computing a disagreement score defined as the variance of each model’s average pairwise discrepancy from the others. If this score exceeds a small threshold $\delta$, we treat the instance as contentious and sample a query with fixed probability $q_t=b/T$.
>
> The KL method triggers annotations when the current loss pattern conflicts with what our belief over model performance would predict. We maintain a posterior belief p_t from past annotations and compare it to a predictive loss distribution $\tilde{p}_t$. The KL divergence is computed as $D_t=KL(\tilde{p}_t||p_t)$ and we query with probability proportional to $D_t​$.
>
> > Each query step requires evaluating  pairwise distances to estimate hypothetical variance (Eq. 8), which could be quadratic in the number of models. What is the computational overhead compared to baselines?
>
> To compute the hypothetical variance at each query step we must evaluate pairwise distances among model outputs. For the LongForm–QA (Canonical) dataset, which contains 54 LLMs (the largest collection of models considered in our experiments), the computational requirement is straightforward to quantify. Given a prompt $x_t$​, each model produces a response $f_i(x_t)$. To estimate the pairwise similarity, we compute the distance $d(f_i(x_t),f_j(x_t))$ for all model pairs. Since the distance is symmetric, i.e., $d(f_i(x_t),f_j(x_t))=d(f_j(x_t),f_i(x_t))$, we only need to evaluate the upper triangular part of the distance matrix excluding the diagonal. The total number of distinct pairwise comparisons is therefore $(54*53)/2=1431$. After computing pairwise similarities, we also compute the oracle similarity for each model output with respect to the annotation $y_t$, namely $d(y_t,f_i(x_t))$ for all 54 models. This adds another 54 distance evaluations per prompt. The total number of distance computations required for a single prompt is thus 1431+54=1485. In practice this computational overhead is negligible. For computational efficiency, we preprocess the entire dataset in advance, computing all pairwise similarities before running the experiments. Running the full pipeline over 1000 prompts on a local machine (Apple MacBook M3 pro, 11-core CPU, 18GB unified memory) yields a total end-to-end time of 2 minutes and 36 seconds. This corresponds to 156 seconds for 1000 prompts, or an average of approximately 0.156 seconds per prompt. Such latency is sufficiently small to support real-time or interactive use cases so the method scales effectively even in settings involving dozens of models.

---

### Meta-Review · Area_Chair_HgWG · 2026-01-07

**Summary:**

This paper studies online active model selection for LLMs under a strict annotation budget, and proposes an algorithm, combining an exponential-weights (Hedge) posterior update with (i) an adaptive learning rate driven by a moving-average variance proxy, and (ii) an active query rule that selects which prompts to label in a stream.

The paper reports broad experiments across 10 datasets and 130+ models, claiming substantial annotation savings and regret reduction.

Across the reviews, there is agreement on several strengths:
+ Practical motivation / well-defined objective: selecting a good model for a stream under limited oracle access is important in deployments
+ Extensive empirical setup (many models, many datasets) and consistent gains over the included baselines
+ Model-agnostic setting

The authors provided a strong rebuttal and clarifies some key misunderstandings in some initial reviews and fills in missing implementation details (for baselines). However, after weighing the full thread, my main concern is that the current submission still leaves some foundational questions insufficiently resolved.

(1) The online setting is simulated via i.i.d. draws from test sets (with repeated realizations), which is reasonable as a first pass but does not reflect certain key challenges of online selection (e.g., distribution shift and temporally correlation) that motivate the problem. The rebuttal acknowledges this and points to the lack of public generative drift benchmarks, but this still leaves the empirical evidence somewhat misaligned with the paper’s claims about online robustness.

(2) Cost model. The algorithm (and baselines) assume that at each time step the system can obtain all candidate models' outputs in order to compute pairwise distances, and then decide whether to query the oracle label. Realistically this assumption can dominate cost and undermine the intended efficiency gain. The rebuttal addresses the distance computation overhead, but does not fully resolve the more fundamental issue that generating all candidate outputs may be the main bottleneck in practice.

(3) Technical novelty (this is subjective so I do not see this as a key limitation for an empirical-foci work) -- The core update is Hedge with importance weighting, and the selection rule uses a plausible variance proxy plus entropy scaling. The paper lacks a sharper argument (theoretical or empirical) for why this particular combination should be expected to reliably better beyond the reported benchmarks, especially given that the adaptive learning rate and active querying rule are acknowledged as heuristic extensions without guarantees.

My recommendation is to reject in the current round, but I see great potential of the paper after strengthening the online evaluation (e.g., controlled shifts in non-i.i.d. streams, even if synthetic) and cost model (e.g., with partial evaluation or candidate pruning).

**Reviewer Concerns:**

Concerns likely addressed
* confusion with routing/MoE
* compute overhead of pairwise-distance calculations

Still outstanding
* limited online evaluation protocol
* requiring all model generations per prompt is a strong assumption
* lack of stronger technical justification

**Reviewer Scores:**

Even with modest upward shift of the scores for all reviewers, I do not expect the post-rebuttal consensus to clearly cross the acceptance threshold

---

### Decision · Program_Chairs · 2026-01-26

Reject